# Norovirus: Facts and Reflections from Past, Present, and Future

**DOI:** 10.3390/v13122399

**Published:** 2021-11-30

**Authors:** Yalda Lucero, David O. Matson, Shai Ashkenazi, Sergio George, Miguel O’Ryan

**Affiliations:** 1Microbiology and Mycology Program, Institute of Biomedical Sciences, Faculty of Medicine, Universidad de Chile, Santiago 8380453, Chile; ylucero@gmail.com (Y.L.); sergio.george@gmail.com (S.G.); 2Hospital Dr. Roberto del Río Hospital, Department of Pediatrics and Pediatric Surgery (Northern Campus), Faculty of Medicine, Universidad de Chile, Santiago 8380418, Chile; 3Clínica Alemana de Santiago, Faculty of Medicine, Universidad del Desarrollo-Clínica Alemana, Santiago 7650568, Chile; 4Eastern Shore Health Department, Virginia Department of Public Health, Accomack County, VA 23301, USA; wrenpt@gmail.com; 5Adelson School of Medicine, Ariel University, Ariel 40700, Israel; shai.ashkenazi7@gmail.com; 6Department of Pediatrics A, Schneider Children’s Medical Center, Petach Tikva 49202, Israel

**Keywords:** Norovirus, epidemiology, prevention, vaccine

## Abstract

Human Norovirus is currently the main viral cause of acute gastroenteritis (AGEs) in most countries worldwide. Nearly 50 years after the discovery of the “Norwalk virus” by Kapikian and colleagues, the scientific and medical community continue to generate new knowledge on the full biological and disease spectrum of Norovirus infection. Nevertheless, several areas remain incompletely understood due to the serious constraints to effectively replicate and propagate the virus. Here, we present a narrated historic perspective and summarize our current knowledge, including insights and reflections on current points of interest for a broad medical community, including clinical and molecular epidemiology, viral–host–microbiota interactions, antivirals, and vaccine prototypes. We also include a reflection on the present and future impacts of the COVID-19 pandemic on Norovirus infection and disease.

## 1. Introduction

For over 50 years now, the scientific and medical community have been gathering knowledge on the full biological and disease spectrum associated with human caliciviruses, which are currently grouped into two genera: the Norovirus and Sapovirus. This review will focus upon Norovirus (NoV), which is currently the most commonly detected human pathogen of the family. We will focus upon key historical aspects and state-of-the-art knowledge that may be of interest to a broad medical readership and we will propose what we believe are the main challenges for better control of NoV-associated disease.

## 2. The Past

### 2.1. Virus Discovery

In 1970, Yow and Melnick, with others [1], reported that no viruses could be imputed to be frequent causes of acute gastroenteritis (AGE) in children. In a several-year study, they, like others, found the bacteria enteropathogenic *E. coli*, *Salmonella* spp., and *Shigella* spp. more often in hospitalized children with AGE than among control children. However, two breakthrough technologies led to the discovery of many novel causes of AGE, including viruses. One such technology was that of the Nobel Prize-winning competition radioimmunoassay of Yalow and Berson [2,3], in which antibody molecules reacted with insulin to form antibody–target complexes that could be quantified. The other, was transmission electron microscopy (TEM) [4], which allowed samples to be magnified up to 70,000 times. On the other hand, at this stage, no animal or cell line model was available able to replicate infection. Thus, research in human volunteers was the only model to advance in characterizing and demonstrating the pathogenic role of this agent in AGE [5,6]. Kapikian was able to visualize NoV particles combining the application of these two technologies, after creating antibody–target complexes (Figure 1) [7]. His original specimens came from an AGE outbreak at an elementary school in Norwalk, Ohio. He studied specimens from people affected in the outbreak and from human volunteers who consented to swallow inoculates from that outbreak. The following is how Kapikian described his original discovery:

“In 1972, a 27 nm virus-like particle was discovered by use of immune electron microscopy (IEM) in an infectious stool filtrate derived from an outbreak of gastroenteritis in an elementary school in Norwalk, Ohio. IEM enabled the direct visualization of antigen-antibody interaction, as the particles were aggregated and coated by specific antibodies. This allowed the recognition and identification of a 27 nm virus-like particle that did not have a distinctive morphology, was low-titered, and was among the smallest viruses known. Serum antibody responses to the 27 nm particle were demonstrated in key individuals infected under natural or experimental conditions; this and other evidence suggested that this virus-like particle was the etiologic agent of the Norwalk gastroenteritis outbreak. The fastidious 27 nm Norwalk virus is now considered to be the prototype strain of a group of non-cultivatable viruses that are important etiologic agents of epidemic gastroenteritis in adults and older children”.[8]

**Figure 1 viruses-13-02399-f001:**
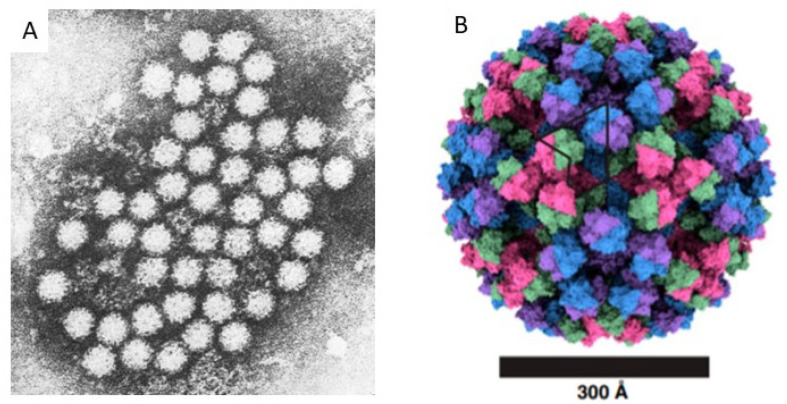
Electron microscopy images of Noroviruses. (**A**). Original image of small round structured viruses, visualized by Kapikian in stool samples from acute gastroenteritis cases, by immune electron microcopy [7]. (**B**). High resolution cryo-electron microscopy of a GII.4 Norovirus particle [9].

Filtered and treated stools administered to human volunteers confirmed that the causative agent was small (<36 nm), nonenveloped, and relatively heat stable [10]. Attempts to in vitro or in vivo passage of the virus, except in human volunteers, were unsuccessful for a long time [11].

### 2.2. The Evolution from Antigenic to Molecular Studies

After Kapikian’s success at virus visualization in 1972, the next major achievement occurred in 1979, when Greenberg and colleagues [12], working with collaborators at the Centers for Disease Control, reported that approximately one-third of outbreaks of AGEs in adults, with no known bacterial or parasitic cause were likely caused by Norwalk-like viruses. This observation was confirmed by many reports in other latitudes.

The first images of NoV provided by Kapikian [7] showed a uniformly round, rough-surfaced particle. Greenberg and colleagues [13] directly assessed proteins from stool specimens containing Norwalk virus and found an approximately 59,000 mW protein, sometimes accompanied by an approximately 30,000 mW protein. These findings were clarified when the genome was sequenced in 1989 by Jiang et al. [14]; one of the open reading frames (ORF) predicted from the sequence encoded a protein with a molecular weight of ~58,000 mW and virus-like particles (VLPs) were created in a baculovirus expression system [15]. The 59,000 mW protein in stool specimens was determined to be the same as the 58,000 protein in the VLPs from the expressed ORF [16]. The virion consists of 180 of these molecules formed in an icosahedron, appearing to be a round structure at resolutions lower than cryo-electron microscopy (Figure 1) [7,9,17]. Further details of the structure are beyond the scope of this review.

As more and more small viruses were characterized, many were recognized to be nearly round, with a single-stranded RNA genome, to have a unique organization of protein-encoding elements, to have a variable role in human and animal disease, and many were found to be causative agents of AGE [18,19,20].

In the 1990’s with the development of enzyme-linked immunosorbent assay (ELISA), further knowledge as to the epidemiologic role of Norwalk-like viruses epidemiology was possible. Paradoxically, these agents were then identified as a major cause of food-borne diarrhea in adults but were infrequently found to be a cause of AGE in children. The largest systematic assessment of hospitalized children found NoV in 7.1% of AGE cases [21]. At the time, it appeared that there may be a difference in pathogenesis between children and adults infected with the same strains, which could explain the differing conclusions. Later, antigenic diversity and the low sensitivity of ELISA tests were determined to be underlying this apparent discrepancy.

Since the 2000s, as genome characterization of many more NoV strains increased, and more investigators shared their information of NoV sequences from different genomic regions, a wide variability became evident and recombination among strains was recognized [22,23], as will be discussed further below.

## 3. The Present

### 3.1. Clinical Epidemiology

Given the progress made in its detection by sensitive molecular diagnostic techniques and the introduction of the rotavirus vaccine, NoV has emerged in recent years as a leading causative agent of AGE in most locations and age groups [24,25,26,27]. It is associated with nearly 20% of all acute diarrheal cases globally, and with an estimated 685 million episodes and 212,000 deaths annually [24,28,29]. Each year in the United States, NoV causes 19 to 21 million illnesses, including 900 deaths, 103,000 hospitalizations, 460,000 emergency department visits, and 2.6 million outpatient visits [30]. It has been identified as a relevant cause of acute endemic diarrhea in children under 5 years of age as well as the primary agent of AGE outbreaks affecting individuals of all ages.

Among 148,867 patients with AGE reported in 178 publications, the overall detection rate of NoV was 17% (95% CI: 15–18%) [31]. In a study comparing 12,531 children with AGE and 11,954 control subjects from 45 countries across the world, the pooled detection rate of NoV was 17.7% (95% CI: 16.3–19.2%) and 6.7% (95% CI: 5.1–8.8%), respectively, giving a pooled OR of 2.7 for the association of NoV infection and AGE (95% CI: 2.2–3.4) [24].

NoV infections are common both in developed and developing countries, affecting all socioeconomic classes; nevertheless, socioeconomic status is related to the risk of acquisition and severity of NoV infections. A systematic review has demonstrated that the detection rate of NoV in AGE episodes decreased from 18% (95% CI: 16–20%) for upper middle-income countries to 15% (13–18%) and 6% (3–10%) for lower middle- and low-income countries, respectively [31]. In another review, NoV was detected in 15% (95% CI 15–16%) of children with AGE in low–middle-income countries (vs. 8% of healthy controls) and 11% (95% CI 10–12%) of those in low-income countries (vs. 9% of asymptomatic controls) [32]. Yet, birth cohort studies conducted in low- and middle-income countries have demonstrated that 66–90% of the children experienced at least one NoV infection and 30–70% experienced NoV-associated AGE in early childhood, with an estimated incidence for NoV-associated AGE of 14 to 66 per 100 child years [25]. Male predominance has been often reported [24,33], with the highest rates of NoV AGE during winter or rainy seasons [34,35,36].

Endemic NoV gastroenteritis typically affects young children under 5 years of age [24,25,37]. A recent meta-analysis found that the highest frequency of NoV infection occurred in infants less than 12 months of age [24]. A second study of hospitalized patients in Indonesia found the highest frequency of NoV infection in those under 24 months of age [34]. In a summary of 10 community-based birth cohort studies, NoV-associated AGE was relatively uncommon in infants under 6 months of age [25].

Foodborne outbreaks are a significant problem for public health, causing outbreaks in otherwise healthy individuals that last 1 to 4 days [26,38]. NoV is the most common cause of such outbreaks, accounting for about 50% of all-cause events worldwide and >90% of viral AGE outbreaks. According to a 2015 World Health Organization report, NoV was the primary cause of foodborne illnesses globally, accounting for 124,803,946 cases and 34,929 deaths, indicating a fatality rate of 0.028 [39]. Similarly, in the United States, NoV caused the highest number of cases of foodborne outbreaks, with an increase from 2563 cases in 1998 to 5135 in 2017, including 4 deaths [40].

Foodborne outbreaks can be caused either by contamination at the source or during food handling, processing, or serving. A systematic review of NoV-attributed outbreaks between January 2003 and July 2017 identified 27 publications that met the definition of confirmed outbreaks caused by food contaminated at source and 47 that met the criteria for outbreaks caused by food handlers [38]. Of all studies, the food type most commonly implicated in outbreaks was seafood (61%); within this group, 89% of outbreaks were associated with oysters and 10% with shellfish. Other commonly infected foods included berries, other fruits or vegetables, salads, and occasionally processed meats [38,40,41]. Outbreaks attributed to food handling involved a diverse range of foodstuffs and occurred in multiple settings, including restaurants (most common), events with catering services such as weddings, and military units [38].

### 3.2. Molecular Epidemiology

The NoV genome comprises a positive-sense, single-stranded, polyadenylated RNA of approximately 7700 nucleotides with 3 open reading frames (ORFs 1–3) [17,24]. Based upon phylogenetic analysis, human NoVs are classified into ten genogroups designated GI to GX, and two tentative genogroups—GNA1 and GNA2—have also been described. Most strains implicated in human disease belong to three genogroups: GI, GII, and GIV [42]. The genogroups are further classified into genotypes based on the capsid protein 1 (VP1) sequence, encoded by ORF2 with more than 45 variants described. Although a consensus genotyping based on VP1 (G-type) is the most widely used, a binomial nomenclature system considering also the RNA polymerase (RdRp) sequence, encoded by ORF1 (P-type), has been proposed, as recombination occasionally occurs at the ORF1/ORF2 junction of the viral genome [42]. In addition, variants (sub-genotypes) are determined by sequence analysis of highly variable regions of the ORF2 and named according to the location and year where they were first described.

Molecular studies have confirmed the complex genomic diversity of circulating NoVs, which vary by location, age, and sometimes the clinical presentation and other variables, with the emergence of new genotypes or variants every few years, that replace previously predominant strains [24,25,31,43,44]. A recent meta-analysis of articles published from January 2015 to May 2020 explored the global distribution of genotypes of NoV infections among children with AGE [24]. Of the 123 studies included in the analysis, which included 120,531 individuals with AGE, most of the NoVs belonged to the GII genogroup (92.9%; 95% CI: 90.6–94.6%), followed by GI, with a significantly lower prevalence (6.7%; 95% CI: 5.2–8.5%) (*p* < 0.0001). A total of 31 genotypes were detected: 12 from the GI genogroup and 19 from the GII genogroup. The most prevalent genotypes were GII.4 (59.3%, 95% CI: 53.4–64.9%), GII.3 (14.9%, 95% CI: 10.6–20.5%), and GII.12 (5.1%, 95% CI: 2.9–8.7%) [24]. The GII.4 strain has consistently been identified as the predominant genotype, with new GII.4 strains emerging every 2–5 years, often replacing previously predominant variants. Between 2000 and 2012, a systematic review of 51 publications identified the following variants as dominant: GII.4/2002, GII.4/2004, GII.4/2006b, and GII.4/2008 [45]. The pandemic GII.4 variant, Sydney 2012, was first reported in early 2012 and soon became the predominant circulating strain globally, replacing those previously described [24,46,47,48].

Selected recent publications on the genotype distribution of human NoVs in various locations and the main variants identified, focused on both sporadic episodes and outbreaks, are detailed in Table 1. These studies differed in various aspects, such as their design (prospective, sometimes population-based, vs. retrospective analysis), age of participants, sporadic cases vs. outbreak-based studies, and/or hospitalized vs. community cases [33,34,43,48,49,50,51,52,53,54,55,56]. Although the relative prevalence of NoV genotypes varied among studies and locations, genotype GII.4 was the most common among all studies. Recombinant strains and GII.4 variants were reported frequently [24,32,33,43,45,46,47,49,54,56] and most of the studies documented changes in the dominant strains during the study period [33,34,35,45,53], with the emerging strain occasionally causing more severe disease [33].

Some differences in the epidemiologic features of the genotypes were observed. For example, in a relatively small study in Qatar, while NoV GII.4 was reported in all age groups, genotype GII.3 infections were more common in children <1 year of age [33]. In Japan, patients infected with genotypes GII.2 and GII.6 were younger than those infected with GII.4 [53]. Geographic differences among circulating genotypes, as well as the viral load associated with disease, were identified. Studies of NoV outbreaks in China showed that GII.2 outbreaks mainly occurred in day care centers, elementary schools, and high schools and were primarily transmitted mainly through person-to-person contact, while GII.4/2012 Sydney outbreaks frequently occurred in colleges and were primarily associated with foodborne transmission [46]. In Japan, outbreaks of the genotype GII.17 occurred frequently as foodborne [53]. Vomiting was more commonly reported by patients infected by GII.2 and GII.17 outbreaks compared with outbreaks associated with the GII.4 genotype [45]. Despite the above, well defined epidemiological or clinical patterns for specific genotypes have not been identified.

The considerable genomic diversity of human NoVs does not necessarily imply antigenic diversity; indeed, antigenic commonality might be of greater benefit if common antigens are neutralizing, as it was demonstrated for rotaviruses (RoVs), for which monoclonal antibodies were created that neutralized several serotypes. As new genetic diversity is described for NoVs, this experience with RoVs ought to be remembered. However, the pathway for NoV antigenic characterization has been much slower and remains a difficult area of study. The major impediment has been inability to readily replicate the virus in the laboratory. Lacking the ability to study specific- and cross-neutralization in cell culture or the human host has been a fundamental impediment to vaccine development. A recent, major advance in NoV studies has been the successful replication of human NoVs in cell cultures [57,58]; yet, to date, the techniques are cumbersome and specific, indicating that widespread application will not likely occur in the near future.

### 3.3. Transmission and Shedding

Human-to-human transmission of NoV is common, mainly by the fecal–oral route (particularly for the epidemic strain GII.4); although, spread may be enhanced by episodes of vomiting [59]. NoV can efficiently survive in the environment and resists freezing temperatures, heating to 60 °C, and disinfection with chlorine or alcohol—facilitating contamination of food, water, and inanimate surfaces [60]—eventually leading to outbreaks, especially in close settings as daycare centers, schools [46,61], hospitals, military camps, and cruise ships [41,62], among others. NoV is highly contagious due to its very low infectious dose; a single particle has an infection probability of 50%, although a dose–response relationship has been noted, in which individuals exposed to higher numbers of viruses experience a higher infectious rate [34,47,61,63].

NoV shedding in stools is maximum during the first 24 to 48 h after symptoms onset, with a mean duration of four weeks [64,65,66]. In immunocompromised hosts, viral shedding in stools can persist for months following infection [65,67].

### 3.4. Clinical Presentation

Outbreaks and volunteer studies have reported about 30% of asymptomatic infections [68]. In symptomatic patients, after a 24–48 h incubation period, a variable combination of acute watery diarrhea, nausea, vomiting, malaise, abdominal pain, and/or headache, that last 1–4 days, have been reported. Fever is less frequent than in RoV infection.

Among children, gastroenteritis due to NoV had slightly lower severity as compared with RoV-caused gastroenteritis, with a 20% decrease in dehydration [69], although a study in Bangladesh found higher rates of moderate–severe dehydration in NoV gastroenteritis [70]. It should be emphasized that older adults (>60 years) are at increased risk of severe and complicated course of NoV AGE. For example, a decade-long analysis of NoV disease risk has shown that older adults living in long-term care facilities in middle–high- and high-income countries experienced frequent NoV outbreaks, leading to high hospitalization rates of 0.5–6% and case fatality rates of 0.3–1.6% [71]. Likewise, of the elderly individuals hospitalized for NoV infection in Atlanta, Georgia (United States), 36% were admitted to intensive care units [44]. Similarly, healthcare-associated infections, mainly in hospital wards, can occur.

Although NoV infections in AGE are mostly short and self-limited, chronic diarrhea in immunocompromised, malnourished, and elderly patients has been described [72,73,74]. On the other hand, immune-mediated disease may be triggered such as necrotizing enterocolitis in newborns [75,76] and inflammatory bowel disease exacerbations [77,78]. Dysregulation of enteric nervous and immune system may also be associated with the development of post-infectious irritable bowel syndrome [79,80].

NoV has also been associated with compromised extraintestinal function and has been detected in serum and cerebrospinal fluid samples [81,82]. Encephalopathy, seizures, and liver disfunction have been described in this context; however, a causal role is still controversial [83,84,85,86].

### 3.5. NoV and Host Interactions

Until recently, when a human NoV (HNoV) replication model was developed in Zebrafish [87], other animal models were not available, and most of the needed evidence about virus–host relationship came from studies of Murine NoV (MNoV) [88]. MNoV can infect and replicate in several immune cells (including T cells, B cells, dendritic cells, macrophages). A specific tropism for Tuft cells, a chemosensory rare type of intestinal epithelial cell (IEC) of CR6 MNoV, which causes persistent infection, has been described [88,89]. MNoV uses a species-specific proteinaceous receptor (CD300LF) to attach cells [90] through binding sites in protruding (P) domains of the major capsid protein (VP1). This relationship is dynamic and dependent on P-domain conformational changes [91,92], which can be induced by cofactors such as bile acids, which enhance cell-attachment [93,94]. Though initial data suggested the binding of MNoV to host carbohydrates [95], recent studies based on nuclear magnetic resonance spectroscopy show that MNoV P-domains do not bind to surface sialoglycans [96].

The detection of viral non-structural proteins in enterocytes of human adults’ biopsies [97], and the presence of viral non-sense RNA in intestinal enteroendocrine cells from pediatric intestinal biopsies [98], suggest that HuNoV replicates in intestinal cells during infection, which is supported by its ability to replicate in human intestinal enteroids (HIE) [57,99]. While replication of HNoV in B cell lines has been shown [100,101], there is lack of evidence of in vivo replication in immune cells [100]. Though a specific protein receptor for HNoV has not been identified, it binds to histo-blood group antigens (HBGAs) [57], carbohydrates expressed on the surface of red blood cells and mucosal epithelial cells that can be also secreted into saliva and other fluids accordingly, when a functional Fucosyltransferase-2 (FUT2) enzyme is expressed (secretor status). Those individuals who do not express a functional FUT2 (non-secretors) are resistant to specific HNoV genotypes [102]. Given the importance of HBGA in HNoV pathogenesis, and as glycoslylation patterns in Zebrafish differs from humans [103], this point should be considered for HNoV in vivo studies, in this model. Additionally, in vitro studies have shown that bile acids can bind to HNoV and promote its replication [104]. The apparently crucial role of gut microbiota in the initial phase of infection is discussed further below.

Once MNoV replicates within host cells, viral molecules are sensed by several innate-immune receptors (e.g., TLR, RIG-1 and MDA5) to promote a type I Interferon (IFN-I) response that is needed to eradicate infection [105]. In humans, both a proinflammatory and a regulatory cytokine response is seen during the acute phase [106], which may be related to symptomatology [107]. HNoV replication in HIEs systems have provided evidence of the importance of innate immunity in its pathogenesis; while HNoV triggers a chemokine (CXCL10) [99] and IFN-stimulated genes with a predominantly type III IFN response; however, there is a strain-dependent sensibility to endogenous IFN which can explain the presence of globally expanded strains more resistant to innate-immunity signaling (such as GII.4) [108]. Adaptive immune response to MNoV requires both humoral and cellular components to be effective [109]. Cellular response is dependent on CD4 and CD8 T cells [110] and requires an effective acute IFN response to be generated, while an ineffective innate and then cellular response has been described in strain CR6, which generates a persistent infection in mice [111]. In the case of HNoV, while the role of antibodies limiting infection has been described [112], data on the cellular response is still limited [113]. However, the ability to cause chronic diarrhea in humans lacking cellular immunity [114] suggests that this component is also necessary to limit the infection.

The association between HNoV infection and later onset of irritable bowel syndrome [80] and inflammatory bowel disease (IBD) [115] has been described. Although evidence from MNoV suggests that infections can trigger IBD in genetically susceptible mice [116], currently there is a gap in our knowledge as to the relevance of this association in humans and the immunological mechanisms involved.

### 3.6. NoV-Gut Microbiota Interactions

A complex relationship between NoV infections and the gut microbiota has been described. Antibiotic-treated mice with subsequent microbiota depletion become resistant to infection with strains associated with both acute (MNV-1) [111] and persistent (CR6) [117] MNoV infections, which suggests the crucial role of the microbiota in promoting infection. Secretory IgA is needed for efficient MNoV infection in mice [118], which is promoted by a functional gut microbiota. Furthermore, antibiotic treatment reduces the number of colonic Tuft cells [119]; this is relevant as the microbiota may play a role in promoting tuft cells; however, this association requires further exploration. HNoV can bind to HBGAs-like molecules in gut bacteria, which stimulates viral replication and resistance to heat stress [120]. On the other hand, the gut microbiota can promote an antiviral host response and limit infection. Specific components of the microbiota, e.g., *Lactobacillus* [121] and poly-γ-glutamic acid from *Bacillus* spp. [122] promote an IFN-I response and limit MNoV infections. The abundance of specific microbiota components (*Ruminococcus* and *Faecalibacterium*) in healthy adults is inversely associated with salivary anti-NoV IgA levels [123], suggesting that these specific bacteria are associated with lower susceptibility to NoV infections. A recently published NoV challenge study in adults showed that the pre-infection microbiota of subjects who developed an asymptomatic infection was enriched in *Bacteroidetes* and depleted in *Clostridia*, compared with the symptomatic subjects [124]. Larger prospective cohort studies are needed to elucidate factors involved in the gut microbiota composition-determined predisposition or protection against NoV infections. Finally, the effect of NoV infection on gut microbiota composition has been a matter of active research. MNoV causes a strain-specific effect on gut microbiota composition, as MNV-1 causes significant alterations [125] not found in persistent CR6 infection [126]. In adult humans, a subset of HNoV-infected adults showed a significant change in gut microbiota characterized by an increase in Proteobacteria and a decrease in Bacteroidetes compared with non-infected controls [127]. In children, viral AGE (including HNoV-caused diarrhea) is associated with changes in gut microbiota, including a decrease in alpha-diversity (which tends to be less significant in HNoV compared with RoV-caused diarrhea), and variable changes in specific taxa according to different studies [128,129,130]. In summary, there is a tripartite relationship between NoV, the host, and the host’s gut microbiota, and details about mechanisms involved, and their potential uses in preventive or therapeutical interventions (e.g., microbiota-related response to vaccines or probiotics) are the subjects of ongoing research.

### 3.7. Antivirals: Not There Yet

Although the traditional NoV therapy has been supportive care, mainly hydration, several antivirals have been explored during the past decades. Virus-directed agents that have been proved included receptor blockers (citrate), protease inhibitors, and nucleoside and non-nucleoside RdRp inhibitors. TLR agonists have been explored to stimulate host immune response. Nitazoxanide has also been tested but its mechanism of action is unknown. These compounds are still in pre-clinical stages. For a comprehensive review in this topic, read [131].

The antiviral would have to not only be safe, but also effective in reducing disease and transmission quickly, because of the short clinical course and high transmission rate of this agent. On the other hand, NoV tends to be under-reported and unrecognized because detection is usually not included in routine workout of AGEs and this point should be improved if an etiological treatment is proposed. Closed environments, such as hospitals, nursing homes for elder individuals, and possibly in military scenarios, where NoV outbreaks could pose significant operational interference, seem ideal first testing sites for candidate antiviral agents.

### 3.8. Path to NoV Vaccine Development

Given that NoV is the leading cause of AGEs globally, that it is associated with significant morbidity and mortality, and that there is no specific anti-viral therapy, preventing NoV infections through vaccination is a high priority [25,132].

However, the development of a NoV vaccine has faced significant challenges. These include the genetic diversity of NoVs with the periodic appearance of new mutants and recombinants, limited knowledge on correlates of protection, the absence of an effective cell line to cultivate the virus, and the absence of an easily used and appropriate animal model. Community-based longitudinal birth cohort studies on NoV infection with genotypic characterization, mainly those conducted in Peru, India, and Bangladesh, shed light on a relatively high (86–100%) homotypic protection after natural infection, namely protection from repeated infections by the same NoV genotype [25]. Along the same line, serum specimens collected from children who had been hospitalized because of NoV AGE contained neutralizing IgG antibodies against homologous GII.4 genotypes, with genotype-specific seroconversion [133]. The duration of protection and the level of heterotypic protections have not yet been fully elucidated.

Several avenues for developing a NoV vaccine have been explored during the past years. Table 2 summaries the current status of these candidates. In summary, the vaccines most advanced in clinical trials—which most likely will “see the light” within the next 5 years—are those based on virus-like particles (VLPs). These candidates, two of which have made it to advanced human trials, have displayed a good safety profile in adults and more recently in infants. The immune responses to these molecules tend to be genogroup- and possibly genotype-specific and it is possible that several genotypes will be required in the vaccine formulation.

## 4. Future

### 4.1. NoV Vaccine Strategies: Coming of Age

Several population groups will eventually benefit from NoV vaccination—older individuals living in nursing homes where outbreaks of NoV gastroenteritis with severe consequences occur, including deaths; adult populations gathered in groups within relatively isolated areas; in strategic functions in which a NoV outbreak could produce a significant disruption (military personnel, peace corps, missions in isolated areas, such as space travel, high altitudes, mines, etc.). In a scenario of more widespread use, travelers may be able to consider a NoV vaccine if the vaccine demonstrated sufficient overall effectiveness in preventing traveler’s diarrhea.

Most importantly, NoV vaccines will be considered for use in children in order to further reduce diarrhea-associated hospitalizations and deaths. A highly effective NoV vaccine has the potential to reduce diarrhea-associated deaths by nearly 200,000 cases per year, with the greatest impact to be seen in the poorest regions of the world [24]. In these underprivileged areas, as well as in more privileged regions, diarrhea-associated hospitalizations could be reduced by nearly 3 million annually.

The implementation of pediatric vaccine programs will have several challenges given current vaccine developments. First, the fact that the most advanced vaccine candidates are to be administered intramuscularly complicates their implementation, as injections require a suitable infrastructure and human resources, that are more complicated and more expensive than those required by oral vaccines. Second, the willingness of parents to “jab” their infants to prevent a diarrheal pathogen in an already crowded schedule of injections, will also be a challenge. In the initial phases of implementation, it seems worthwhile to consider a NoV jab somewhere between 6 and 12 months of age, in order to not continue to crowd the 0–6 months schedule. The fact that NoV infections peak after 6 months of age could favor such an approach. Alternatively, and likely the optimal course of action, would be to include NoV in combination with another respiratory and/or enteric pathogen, for example Rotavirus. One common vaccine for two main diarrheal pathogens seems to be a reasonable avenue for the future. Data from Rotavirus suggest that both gut microbiota and host factors (e.g., secretor status) can influence the host’s response to a vaccine, and should be carefully assessed especially if a combined Rotavirus–NoV vaccination strategy is implemented.

### 4.2. Lessons from the “Coronavirus Pandemic”

The coronavirus pandemic has had important “indirect effects”, including an impressive reduction in respiratory and gastrointestinal infections throughout 2020 and at least the first half of 2021. In most countries with active respiratory virus surveillance, the “traditional” autumn–winter epidemic surge of respiratory syncytial virus (RSV), influenza, and other respiratory viruses was completely blunted [154]. For gastrointestinal viruses, epidemiological surveillance is in generally incomplete; nevertheless, a similar phenomenon was observed, particularly for NoV [155,156,157,158,159,160]. Several hypotheses can be generated to explain this significant decrease, including (i) school closures, leading to massive reductions in gatherings of children and children with adults; (ii) prolonged and intense quarantines; (iii) massive increases in hand hygiene procedures; (iv) global masking; (v) reduction in enteric pathogen testing capacity; and (vi) possible secondary effect of SARS-CoV2 infection (which can replicate on the gastrointestinal epithelium) on the predisposition to other enteric viruses, e.g., altering the gut microbiota composition or inducing an antiviral IFN response [157,160].

Thus, in what could be considered a natural global experiment, society has learned that mucosal pathogens, respiratory and enteric, causing significant disease every year, could eventually be controlled with important societal “restrictions”. Understandably, such restrictions have significant negative impacts, particularly school closures, that make them unfeasible in a non-pandemic situation. However, we can learn from them and partially apply measures that could aid in better control of these mucosal pathogens. Avoiding overcrowding in school classrooms, possibly by increasing and perfecting hybrid, presential/non-presential sessions, especially during harsh winter peaks in infections, could be a future consideration. Temporal, partial, or full school closures to control epidemic surges should also be on the list of possibilities. Hand and respiratory hygiene measures should remain and be enforced overtime, as there will be a natural tendency to relax these measures as the pandemic is forgotten. The regular use of face masks in reasonably eligible school aged children and adults will be a matter of discussion and debates in the near future. The potential positive effects of a reduction in pathogens spread through nasopharyngeal droplets will have to be weighted with the negative psychological effects of face covering, especially during the childhood years. Rigorous studies will be required to address these issues and provide evidence for reasonable future recommendations.

Resurgence of norovirus incidence after relaxation of non-pharmaceutical interventions related to SARS-CoV2 pandemic has been predicted [161]. Continued surveillance should be encouraged to allow adequate preparation for a potential increase in healthcare pressures beyond previous levels.

## Figures and Tables

**Table 1 viruses-13-02399-t001:** Selected recent publications (2018–2021) of the genotype distribution of human noroviruses.

Year Published	Years of Study	Location	NO. of Patients	Genogroups/Genotypes (%)	Comments	Reference
2021	2015–2020	Global	120,531	GII.4 (51)	Meta-analysis	[24]
				GII.3 (15)		
				GII.12 (5)		
2021	2010–2016	Brazil	251	GII.4[P31] (64)		[49]
				GII.17[P17] (6)		
				GII.1[P33] (6)		
2021	2017–2018	China	1500	GII.4 (44)	Outbreaks	[50]
				GII.17 (27)		
2020	2016–2017	Chile	174	GII.4 (35)	Surveillance Recombinants	[51]
				GII.6 (23)		
				GII.7 (12)		
2020	2015–2019	Indonesia	966	GII.[P31] (44)	HC	[34]
				GII.[P16] (37)		
2020	2009–2013	China	3134	GII.4 (50)	HP	[48]
				GII.17 (11)		
				GII.3 (8)		
2019	2014–2018	Brazil	61	GII.4 (19)	<5 years	[43]
				GII.6 (19)		
				GII.7 (19)		
2019	2016–2018	Qatar	177	GII.4 (62)	Children	[33]
				GII.2 (16)		
				GII.3 (14)		
2019	2014–2018	China	3422	GII.4 (72)	Outbreaks	[46]
				GII.3 (14)		
				GII.17 (8)		
2019	2010–2012	Bangladesh	819	GII.4 (33)	HC, <5 years	[52]
				GII.3 (13)		
				GII.6 (11)		
				GII.13 (11)		
2019	2012–2018	Japan	4588	GII.4 (22)	Surveillance	[53]
				GII.2 (15)		
				GII.17 (6)		
				GII.6 (4)		
2018	2015–2017	Thailand	1591	GII.4 (32)	Surveillance	[54]
				GII.17 (12)		
2018	2013–2015	Botswana	484	GII.4 (70)	HC, <5 years	[55]
				GII.2 (9)		
				GII.12 (9)		
				GI.9 (7)		
2018	2012–2013	Angola	343	GII.4 (20)	<5 years	[56]
				GII.6 (15)		
				GI.3 (12)		
				GII.10 (10)		

Abbreviations: HP—hospitalized patients; HC—hospitalized children.

**Table 2 viruses-13-02399-t002:** Vaccine candidates against Norovirus.

Vaccine Platform	Specific Antigens in the Vaccine	Expression System	Route and Schedule of Administration	Stage of Development	References
Virus-like particles	Bivalent GI.1 and GII.4	Baculovirus system	First delivered by intranasal route, and currently developed for intramuscular administration. Two doses separated by 30 days.	Clinical: Phase IIb completed; advancing to phase III trials in adults and children.	[134,135,136,137,138,139]
	Bivalent GI.1 and GII.4	Hansenula polymorpha	Intramuscular administration. Two to three doses separated by 28 days.	Clinical: Phase I completed; advancing to Phase II trials.	[139,140]
	Quadrivalent GI.1, GII.3, GII.4, GII.17	Pichia pastoris system	Intramuscular administration; doses under evaluation.	Clinical: Phase I/IIa ongoing	[139,141,142]
	Monovalent GII.4 VLPs	Plant expression system (tobacco, potato)	Oral and intranasal administration	Pre-clinical	[139,143,144]
	Quadrivalent GI.1, GI.3, GII.4, GII.12	Baculovirus expression system	Intramuscular administration.	Pre-clinical	[139]
	Trivalent Norovirus GI.3 and GII.4 and Rotavirus rVP6	Baculovirus expression system	Intramuscular administration.	Pre-clinical	[139,145,146]
	Bivalent Norovirus GII.4 and Enterovirus 71	Baculovirus expression system	Intraperitoneal administration.	Pre-clinical	[139,147]
P-particles	Monovalent GII.4	Baculovirus expression system	Intranasal administration.	Pre-clinical	[139,148]
	Monovalent GII.4 enhanced by adjuvant FlaB.	*E. coli* system	Intranasal and sublingual administration.	Pre-clinical	[139,149]
	Trivalent Norovirus, Hepatitis E and Astrovirus	*E. coli* system	Intranasal administration.	Pre-clinical	[139,150]
	Monovalent GII.2	Viral replicons in eukaryotic cell lines	Intranasal administration.	Pre-clinical	[139,151,152]
Adenovirus vector-based	Monovalent GI.1 or GII.4; or Bivalent GI.1 and GII.4 (co-expression with a double-stranded RNA adjuvant.	Human host cells	Oral administration; doses under evaluation.	Clinical: Phase I in adults completed; advancing to Phase II trials.	[139,153]

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
