# Peer review of "Norovirus: Facts and Reflections from Past, Present, and Future"

_viruses, 2021, doi:10.3390/v13122399_

Round 1
Reviewer 1 Report
This review of noroviruses by Lucero et al. provides an overview of the history, current understanding, and future directions of the field. Overall, the review is well-written. The main area of improvement would be additional updated references for the human noroviruses. The following suggestions seek to improve the manuscript:
- Consider editing the sentence starting on Line 46: “One ethical standard also was somewhat different from what it is permissible today; that is the ready participation…”. Considering that volunteer challenge studies have continued to be utilized even to the present day and must undergo rigorous scientific review and safety testing of inocula, this statement might imply unintentionally that they are not ethical. If the point is “ready participation” that may also need rewording because recently published volunteer studies have been able to recruit adequate numbers of participants.
- Highlighting the development of competitive radioimmunoassays was interesting (line 42), but adding a specific reference here relating to AGE viruses would give context for the reader.
- The section on target cells focuses primarily on murine noroviruses. However, it would be important to include work about human norovirus target cells in the gut as well. Karandikar et al., first showed evidence for norovirus replication in epithelial cells in intestinal biopsy studies https://pubmed.ncbi.nlm.nih.gov/27412790/ Later, Green et al. showed evidence for replication of human norovirus in intestinal epithelial cells including enteroendocrine cells. https://www.nature.com/articles/s41467-020-16491-3
Although comparisons are made to murine norovirus, there are likely important differences in their targeting of immune cells. Neither study showed evidence for active replication of human norovirus in immune cells. The human biopsy data is consistent with the replication of certain human noroviruses in enteroid cell cultures, which are primarily epithelial in composition. https://www.science.org/doi/full/10.1126/science.aaf5211
Reviewed here: https://www.ncbi.nlm.nih.gov/pmc/articles/PMC6669637/
- Line 280: Is the IEC abbreviation referring to “intraepithelial” lymphocytes or “intestinal” epithelial cells? The next sentence refers to tuft cells as an IEC, so this distinction is important since tuft cells are a class of epithelial cells (not intraepithelial lymphocytes). Please clarify.
- Suggest updating review with some recent data regarding human noroviruses in the enteroid system
https://www.pnas.org/content/117/38/23782
https://pubmed.ncbi.nlm.nih.gov/33494515/
- There are published studies emerging about the effect of the coronavirus pandemic on norovirus that support the authors ideas, such as this paper by Kraay et al.:
https://pubmed.ncbi.nlm.nih.gov/33606027/
Minor editing comments:
- Is the word “facts” needed in the title of the article? A suggestion: Norovirus: Past, Present, and Future
- Genus names should be capitalized and written in italics (line 30)
- Line 85: would be consistent with rest of paragraph to note the first authors name for genome sequencing, a key advance in the history
- Line 293, “interferon” in this journal (not interpheron)
- Group A Rotaviruses are the dominant rotaviruses associated with diarrheal disease and the abbreviation “RVA” is a more common abbreviation than RoV. Line 113, can change to: Rotavirus A (RVA) vaccine. Alternatively, can write out “rotavirus” in paper.
- Line 238: “is good at” could be reworded
- Table 1: Could title the fifth column as Genogroup/Genotype (%). This would allow streamlining by removing the percent sign next to numbers throughout the column.
Author Response
Dear Reviewer,
We thank your comments and suggestions. Please see our reply in the attached file.
Kind regards,
Yalda.

Reviewer 2 Report
The manuscript by Lucero et al. is a well performed work that aims to summarize our understanding on human and murine noroviruses. The review is well conducted and the points are well defined as well as the adequate number of figures and tables and their content. I have however some points which require correction prior publication:
Mayor issues:
1. 277-278: “Given that Human NoV (HNoV) cannot replicate in animal models, the use of Murine NoV (MNoV) has provided needed evidence about virus-host interaction”
The autors seem to forget the development of Zebrafish larvae as an animal model for replication of human NoV. See: https://doi.org/10.1371/journal.ppat.1008009 and https://doi.org/10.1038/nprot.2013.068.
It may also be worth mentioning that the glycosylation in Zebrafish differs from that in human (https://doi.org/10.1038/s41467-018-06950-3)
2. 280-283: “A specific tropism for Tuft cells, a chemosensory rare type of IECs, of CR6 MNoV which causes persistent infection, have been described (86). MNoV uses a species-specific proteinaceous receptor (CD300LF) to attach cells (87), and may also use carbohydrates as attachment receptors with varied specificity according to strains (88)”.
This claim is unfortunately based on outdated bibliography. At that time, S. Taube and co-workers (88) extracted their conclusions from neuraminidase treatment of cell cultures and from directed mutagenesis. Enzyme treatment of cell cultures affect cells metabolism in unpredicted ways, and the mutations generated in this work are located in the region where the receptor CD300lf binds. Interaction of the outmost region of VP1 protein from MNoV with the receptor also depends on GCDCA and metal binding, and requires reorientation of local loops. It is therefore highly likely that mutagenesis altered receptor recognition. For more details see: https://doi.org/10.1073/pnas.1805797115, https://doi.org/10.1128/JVI.00970-19, https://doi.org/10.3390/v11030235 and https://doi.org/10.1371/journal.pbio.3000649.
The groups of S. Taube and T. Peters have recently put considerable effort into the study of the interaction between carbohydrates and mNoV at atomic resolution. Systematic NMR experiments have unequivocally shown that there is no binding between MNoV protruding domains and HBGAs or gangliosides (https://doi.org/10.3390/v13030416). This is in line with the absence of crystal structures of carbohydrates bound to MNoV. Therefore, I would suggest to rephrase the sentence and add the missing bibliography.
3. The style of the references is inconsistent with the guidelines given by the Journal. It is specially noticeable in the names of authors. For example, reference 9 shows the correct style, whilst reference 10 uses only the firs letter for every name and surname. The use of capital letters is also inconsistent. It is also noticeable that the links to online available references not always direct to the original journal. For example, reference 11 directs to Pubmed. Another example is ref. 88, which does not direct to the article itself. Why not use the DOI instead? DOI is the most widely used article identifier, and directs to the cited article when used in the following form: https://doi.org/xxxxxxxxx.
Minor:
mW: The most common abbreviation for molecular weight is MW. mW may be misleading.
Fig. 1: The footnote describes figures A and B, but not such letters appear in the figure. Also, the right caption shows a letter E probably from the original publication, which should be removed. Can the authors use higher resolution pictures?
20, 21 and 23: The abstract mentions 3 times the word include/including. It is just a matter of style, but considering that the abstract condensate the essence of the article, I would suggest to polish it by using a synonym.
28: “medical community has been..:”
106-109: Lengthy paragraph. I recommend to split into 2 sentences.
170: There also two tentative genogroups, namely GNA1, GNA2.
233: yet to date … to date. Please correct.
245: Infectious rate instead of attack rate.
246: A style recommendation: Perhaps “maximum” instead “greatest”?
310: Use relationship instead of interaction?
Author Response
Dear Reviewer,
We thank your enlightening comments and suggestions. Please see our reply in the attached file below.
Kind regards,
Yalda.

Round 2
Reviewer 1 Report
The authors have responded well to both reviewers, and the manuscript is improved.